# Measurement of Cloud Top Height: Comparison of MODIS and Ground-Based Millimeter Radar

Juan Huo [1,2,*], Jie Li [2,3], Minzheng Duan [1,2], Daren Lv [1,2], Congzheng Han [1,2] and Yongheng Bi [1]

[1] Key Laboratory of Middle Atmospheric and Global Environment Observation, Institute of Atmospheric of Physics, Chinese Academy of Sciences, Beijing 100029, China; dmz@mail.iap.ac.cn (M.D.); ludr@mail.iap.ac.cn (D.L.); c.han@mail.iap.ac.cn (C.H.); byh@mail.iap.ac.cn (Y.B.)

[2] University of Chinese Academy of Sciences, Beijing 100049, China; lijie8074@mail.iap.ac.cn

[3] State Key Laboratory of Atmospheric Boundary Layer Physics and Atmospheric Chemistry, Institute of Atmospheric of Physics, Chinese Academy of Sciences, Beijing 100049, China

* Correspondence: huojuan@mail.iap.ac.cn

**Abstract:** Cloud top height (CTH) is an essential pareter for the general circulation model in understanding the impact of clouds on the Earth's radiation budget and global climate change. This paper compares the CTH products, derived from the Moderate Resolution Imaging Spectroradiometer (MODIS), onboard the Aqua and Terra satellites with ground-based Ka band radar data in Beijing from 2014 to 2017. The aim was to investigate the data accuracy and the difference in CTH measurements between passive satellite data and active ground-based radar data. The results show that MODIS, on average, underestimates CTH relative to radar by $-1.08 \pm 2.48$ km, but with a median difference of $-0.65$ km and about 48% of differences are within 1 km. Statistically, MODIS CTHs which are greater than 6 km show lower discrepancy to radar CTH than those of MODIS CTHs less than 4 km. The CTH difference is independent of cloud fraction and cloud layer. It shows strong dependence on cloud depth, decreasing as cloud depth increases. There is a tendency for MODIS to underestimate high thin clouds but overestimate low thin clouds relative to radar. Total ozone, $SO_2$, CO, $NO_2$, aerosol $PM_{10}$, total water vapor and temperature inversion show unobvious influences in the CTH discrepancy. It is shown that the MODIS $CO_2$-slicing technique performs much better than IRW (infrared window) technique when cloud layer is higher than 2 km. The average difference calculated from all comparisons by $CO_2$-slicing technique and IRW technique is $0.09 \pm 1.58$ km, and $-2.20 \pm 2.73$ km, respectively.

**Keywords:** cloud top height; MODIS; radar; accuracy; validation

## 1. Introduction

Clouds cover more than 50% of the globe. They play important roles in regulating the energy budget and hydrological cycle of the Earth–atmosphere system [1–4]. However, clouds are one of the least-understood components, as well as being one of the largest uncertainty sources, in general circulation model (GCM) simulations [5–7]. Cloud top height (CTH) is a fundamental parameter, describing the vertical distribution of clouds, which partly determines whether clouds exert a warming or cooling effect on the planetary radiation budget.

MODIS measures the infrared (IR) brightness temperature of the cloud to derive CTH and most passive satellite imaging instruments in operation are using similar method to derive CTH [8–13]. Since its launch in 2000, MODIS onboard the Aqua and Terra satellites has provided long-term

CTH products containing a high volume of data and these datasets are being widely used by the meteorological community [2,11,12,14–24]. Alongside improvements in the CTH retrieval algorithm, investigations into the uncertainties of the CTH products of MODIS have been performed. For example, Holz et al. carried out a detailed evaluation of global MODIS cloud products using two months of Lidar observations on the CALIPSO platform, and a global mean difference of −1.4 ± 2.9 km was reported [25]. Garay et al. compared MODIS CTH with ship-based observations in the southeastern Pacific and found MODIS retrievals were biased high by more than 2 km [26]. Xi et al. compared the marine boundary layer CTH derived from CERES-MODIS with ARM radar-lidar measurements in the Azores and a 0.063 km bias was found [22]. After the Collection-6 MODIS CTH products were released in 2014, Håkansson et al. compared them with CloudSat-CPR (Cloud Profiling Radar) data and reported that the global mean difference was −0.61 ± 2.53 km [18].

Millimeter wavelength radar can penetrate the interior of clouds and is a powerful method for observing the macroscopic and microphysical properties of clouds. The CTH can be measured directly by radar returning echoes. Therefore, the radar provides a reliable approach in measuring the CTH, and an effective approach in investigating the reliability of the CTH retrieval algorithm, which was developed for passive remote sensors [27–31].

In MODIS CTH retrieval algorithm, the atmospheric profiles input for the clear-sky radiance calculation affect the retrieval accuracy because their differences with the actual profiles determine the calculation accuracy directly. To be a globally adaptive algorithm, MODIS applies a latitudinal clear-sky radiance bias adjustment in mitigating the difference between observed and calculated radiance [9]. However, such an adjustment, calculated from historical data, might be too slow for capturing real-time local weather properties, and a latitudinal adjustment is too coarse for regions greatly affected by human activities, resulting in uncertainties with regional preferences.

The aim of this paper is to investigate the difference of the MODIS CTH products with ground-based radar data obtained in Beijing, over a long-term period, and to investigate its relationship with the cloud properties and atmospheric parameters. Beijing (39.96°N, 116.37°E) is located in northern China, has a continental monsoon climate and four distinct seasons. The dry and cold winter is governed by the northwest monsoon from the continent, while the wet and hot summer is dominated by the southeast monsoon from the ocean. Spring and autumn are the transitions between these two monsoons. Air pollution events have begun to occur more frequently, owing to the development of industry. Diverse cloud distributions and complex atmospheric conditions challenge the MODIS CTH retrieval algorithm. Investigating the difference under various and complex conditions benefits a thorough understanding of the accuracy of the MODIS CTH products. Baum et al. stated that "A mission of the National Aeronautics and Space Administration (NASA) Earth Observation System is to provide long-term data products of superior accuracy and reliability from sensors on the Aqua and Terra platforms" [14]. Thus, investigating the accuracy in a diverse range of regions comprehensively is a necessary step for establishing cloud product datasets of climate quality and climate data records. In addition, comparison results provide references for the application of MODIS data in climate models and the improvement of the retrieval algorithm.

The rest of this paper is organized as follows: Descriptions of the MODIS data, radar data and other measurement data of atmospheric properties are provided in Section 2. An overview of the distribution of MODIS CTH and Ka radar CTH and their differences is given in Section 3. Section 4 presents a detailed analysis of the influences of cloud properties and atmospheric parameters on CTH uncertainties. Conclusions and some discussion points are given in Section 5.

## 2. Materials and Methods

### 2.1. MODIS CTH Retrieval Algorithm

MODIS is a key instrument on the Aqua and Terra satellites that scans the entire globe every 1–2 days (NASA's Earth Observing System Home Page. Available online: https://eospso.nasa.gov//;

accessed on 18 May 2020). It measures radiance in 36 spectral bands, covering the spectral range from 0.42 to 14.24 µm, at three spatial resolutions (250, 500 and 1000 m). The swath dimensions are 2330 km (cross-track) by 10 km (along-track at nadir) (MODIS Home Page. Available online: https://modis.gsfc.nasa.gov/; accessed on 18 May 2020). The MODIS CTH retrieval algorithm has undergone several updates. The latest version (Collection 6) is a combination of the CO2-slicing technique (also known as the radiance rationing technique) and the 11-µm infrared window technique (IRW technique).

The IRW technique is based on the assumption that clouds are black bodies, i.e., the emissivity of cloud is assumed to be 1 [32,33]. The measured 11-µm infrared brightness temperature is regarded as the brightness temperature of the cloud top, which can be written as,

$$R_\lambda = B_\lambda(T_c),$$ (1)

where $\lambda$ is wavelength, $R_\lambda$ is the measured brightness temperature and $B_\lambda$ is the brightness temperature of a black body at the temperature of $T_c$. The measured 11-µm brightness temperature is used to compare with the brightness temperature profile derived from a gridded temperature profile product, that is provided by the National Centers for Environmental Prediction (NCEP) Global Forecast System, using a radiative transfer model called the Pressure Layer Fast Algorithm for Atmospheric Transmittances (PFAAST) [34]. The MODIS CTH is then provided by the match in brightness temperature.

The $CO_2$-slicing technique has a long history [9,35–37]. A detailed description is presented in Menzel et al. [9], so only a brief account is given in the present paper. Equations (2)–(4) present the core principles,

$$R_v - R_{clr}(v) = NE[R_{bcd}(v, P_c) - R_{clr}(v)],$$ (2)

$$R_{bcd}(v,\ P_c) = R_{clr}(v) - \int_{P_c}^{P_s} \tau(v,p) \frac{dB[v,\ T(p)]}{dp} dp,$$ (3)

$$\frac{R(v_1) - R_{clr}(v_1)}{R(v_2) - R_{clr}(v_2)} = \frac{NE_1 \int_{P_s}^{P_c} \tau(v_1,\ p) \frac{dB[\ v_1, T(p)]}{dp} dp}{NE_2 \int_{P_s}^{P_c} \tau(v_2,\ p) \frac{dB[v_2,\ T(p)]}{dp} dp},$$ (4)

where $v$ is the frequency, $R(v)$ is the measured radiance, $N$ (0~1) is the cloud coverage of the field of view (FOV), $E$ is the emissivity of the clouds, $R_{clr}$ is the clear-sky radiance, $R_{bcd}$ $(v,P_c)$ is the opaque cloud radiance from pressure level $P_c$, $\tau(v, p)$ is the fractional transmittance of radiation at frequency $v$ from the atmospheric pressure level $p$ arriving at the top of the atmosphere ($p = 0$), $P_c$ is the cloud-top pressure, $B(v, T(p))$ is the Planck radiance at wavelength v at temperature $T(p)$, and $P_s$ is the surface pressure. For a cloud element in a FOV, Equation (2) describes the relationship between the observed radiance $R(v)$ and cloud radiance. The radiance from cloud can be expressed by Equation (3). Equation (4) is derived from Equation (2) and Equation (3). For two neighboring wavelengths, located within the 15-µm $CO_2$ absorption region, the differences in the emissivity are assumed to be negligible (i.e., $E_1 \approx E_2$). The CTH can be derived from the ratio by comparing it with the model calculation input, with the atmospheric temperature profile and transmittance profile (NCEP-PFAAST), at two nearby wavelengths.

In the Collection-6 retrieval algorithm, the $CO_2$-slicing technique is limited to ice clouds (mid- to high-level clouds), which are identified according to the cloud emissivity ratios. When the difference between cloud radiance and clear-sky radiance is so small that the $CO_2$-slicing technique is unsuitable to retrieve the CTH, MODIS uses the IRW technique. The MODIS CTH products used throughout this paper are the Collection-6 1 km CTH products (MYD06/MOD06) from the Aqua and Terra satellites [38].

### 2.2. Ka Band Radar

The Ka band polarization Doppler radar (KPDR; wavelength: 8.55 mm), located at the Institute of Atmospheric Physics (IAP), Beijing, China (39.967°N, 116.367°E), was set up in 2010 [39]. The primary technical specifications of KPDR are presented in Table 1. KPDR was set to work 24 hours every day, mostly in a vertically pointing mode. In 2013, 2018 and 2019, KPDR was taken offline during almost the whole of the summer period. Therefore, this paper uses the radar data collected from 1 January 2014 to 31 December 2017 to compare with MODIS data. The effective vertically pointing observation time accounted for more than 85% of the four years.

**Table 1.** Technical specifications of the KPDR at the IAP, Beijing, China.

| Parameter | Technical Specification |
|---|---|
| Frequency | 35.075 GHz |
| Peak power | 29 kW |
| Pulse length | 0.2 µs |
| Type | Magnetron |
| Diameter | 1.5 m |
| Gain | 54 dB |
| Scanning mode | Vertically pointing, plan position indicator, range–height indicator |
| Beam width | 0.4° |
| Vertical resolution | 30 m |

KPDR uses a magnetron transmitter, which transmits stronger signals than an all-solid transmitter, and a threshold of −45 dBz is used for cloud identification. A data-quality control algorithm, using the threshold method and median filter method, is applied to eliminate the non-atmospheric reflectivity or clutter [40]. For a radar cloudy profile, cloud top height is the height of the top cloudy bin. The vertical resolution of KPDR is 30 m and the CTH uncertainty of a radar cloudy profile is 30 m theoretically. The KPDR CTH (expressed as *H*) of a single layer cloud containing large number of continuous cloudy profiles is the mean top height of all cloudy profiles. Two methods are used to calculate the KPDR CTH for multi-layer clouds. One method is the mean top height of all cloudy profiles, which is similar with single layer cloud (also expressed as *H*); the other method is the average CTH of all profiles from the top-level cloud (expressed as *H'*). In Section 4.1.4, contrast between the *H* and *H'* is presented which illustrates very close statistics.

### 2.3. Other Data

As expressed in Equations (1) to (4), the errors of the MODIS CTH retrieval algorithm can be partly produced by methods, including calculating the uncertainties from the radiative transfer model, the inaccurate descriptions of atmospheric parameters. This paper focuses on quantifying the CTH differences between MODIS and KPDR, analyzing the relationships of these differences with the physical properties of cloud and investigating impacts of local atmospheric parameters on CTH differences.

Part of the cloud properties and atmospheric properties data, including the cloud fraction, temperature, total water vapor (TWV) and total ozone data, are from the ERA5 dataset of the ECMWF (European Centre for Medium-Range Weather Forecasts; http://apps.ecmwf.int/datasets/; accessed on 18 May 2020). In addition, the surface $PM_{10}$ (coarse particulate matter), $NO_2$, CO and $SO_2$ data from synergistic measurements at a site within 3 km of the IAP, recorded by the China National Environmental Monitoring Center (China National Environmental Monitoring Center; http://106.37.208.233:20035/; accessed on 18 May 2020), are used to investigate their influences on CTH uncertainties.

*2.4. Data Collocation Method*

KPDR scans three profiles every second with a beam width of 0.4°. The MODIS onboard the Aqua and Terra satellites passes over the IAP, roughly twice a day. The 1 km spatial resolution is applicable for the nadir area but inapplicable for other areas owing to the changed MODIS viewing geometry. The spatial and temporal resolution of the KPDR and MODIS data are different and therefore a data collocation approach is necessary. Naud and Muller averaged MODIS CTH data over a ± 0.1 latitude–longitude box and compared these data with the maximum CTH measured simultaneously by a surface W-band radar system [21]. Dong et al. compared surface data averaged over a 1 h interval with satellite data averaged within a 30 km × 30 km area [16]. Chang et al. compared aircraft lidar CTH data averaged over a 10 s interval with satellite measurements [15]. These collocation methods are designed in different ways according to the observation conditions, which provide us with references for data collocation. In order to minimize the discrepancies resulting from the spatial and temporal sampling differences, four collocation methods were contrasted and the optimal one obtained, as presented in Appendix A. Based on the optimal collocation scheme that the MODIS CTH averaged within 5 km is compared with the mean KPDR CTH within 10 min (± 5 min), the CTH difference investigations should provide reliable results. The KPDR cloud base height (CBH) is the mean base height of all cloudy profiles, and cloud depth is equal to the CTH minus the CBH.

## 3. Results Overview of CTHs Derived from MODIS and KPDR

From 2014 to 2017, about 4000 valid MODIS CTHs (> 0) are obtained from the area within 5 km, and 30 km of the IAP, respectively. The monthly distributions of CTHs, derived from MODIS and KPDR, are presented in Figure 1. The value of the KPDR CTH, hereinafter, is termed $H_r$ and the value of MODIS CTH is termed $H_m$. It is shown that all of the monthly $H_m$ values, including those averaged in the 5 km area and 30 km area, are lower than the monthly $H_r$ values. In each month, the mean $H_r$ is greater than 6 km, and summer (June to August) has a higher CTH than other seasons. The mean $H_m$ in each month is lower than 6 km and decreases by 1–3 km relative to $H_r$. The standard deviation (STD) of $H_m$ is higher than that of $H_r$, indicating a larger range of variation. A raw comparison based on monthly statistics shows that there is an obvious discrepancy (>2 km) between $H_r$ and $H_m$. One reason behind such a discrepancy is that the temporal resolution of the two datasets is different. Besides, it will also be associated with the uncertainties in CTH retrieval.

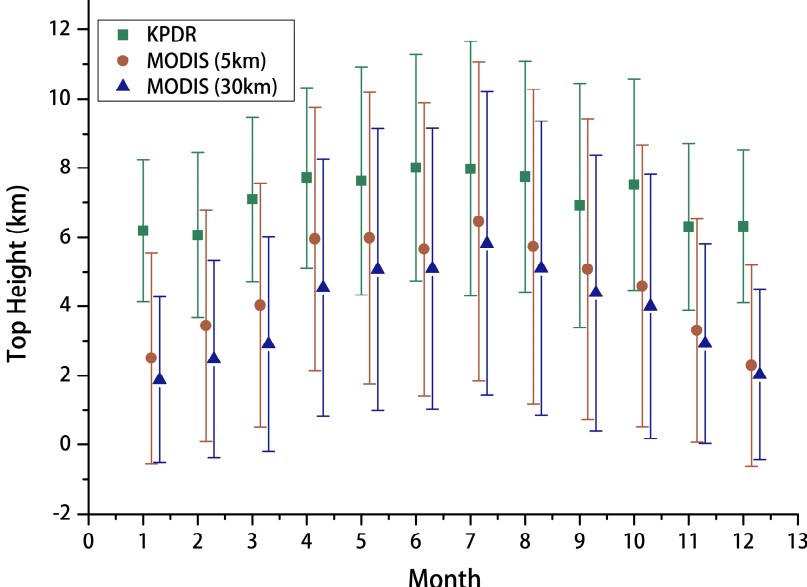

**Figure 1.** Monthly $H_m$s from KPDR data (squares) and MODIS data averaged within a 5 km (circles) and 30 km (triangles) area of the IAP, Beijing and the corresponding STDs.

As shown in Figure 1, the mean $H_m$ within 5 km is closer to the $H_r$ than the $H_m$ within 30 km, indicating that the collocation scheme using a 5 km radius should be better for collocating the MODIS CTH data with the KPDR CTH data than with a 30 km radius. Based on the 5 km collocation scheme, all effective comparison collocations (about 2300) are presented in Figure 2a. The distribution of the difference between MODIS and KPDR ($D_{mr}$, equal to $H_m$ minus $H_r$) is shown in Figure 2b.

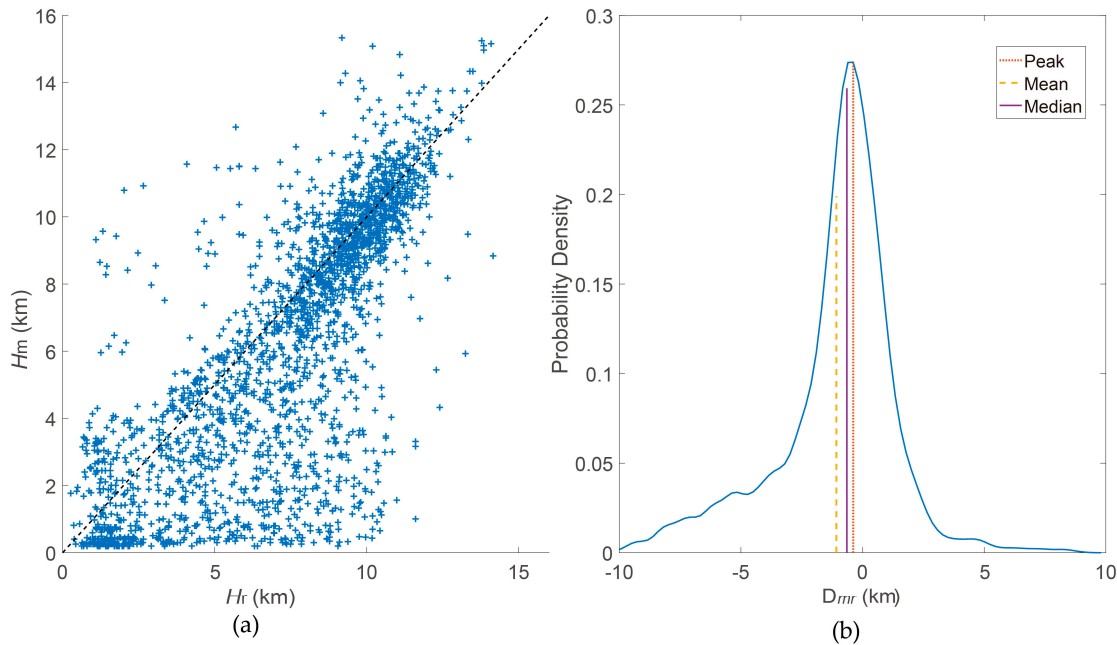

**Figure 2.** All CTH comparisons collocated from MODIS and KPDR (**a**) and the distribution of their differences $D_{mr}$ (**b**). The mean $D_{mr}$ and median $D_{mr}$ are indicated with different lines.

Figure 2b illustrates a non-Gaussian distribution of the $D_{mr}$. The mean difference (mean), STD, median, inter-quartile range (IQR), the difference with a peak probability density (termed 'peak'), and the mean absolute difference (termed 'model') are calculated and presented in Table 2. The MODIS CTH is, in general, lower than the KPDR CTH. There are about 62% of cases of $D_{mr}$ within 1.5km, 48% within 1 km, 25% within 0.5 km, and 13% within 0.25 km. It is found that the difference declines greatly when $H_m$ is higher than 6 km (see Figure 2a). Comparisons show that $H_m > 6$ km has closer values to $H_r$ than $H_m < 4$ km. In other words, $H_m > 6$ km is more accurate than $H_m < 4$ km.

**Table 2.** $D_{mr}$ statistics from all comparisons, comparisons of $H_m < 4$ km, and comparisons of $H_m > 6$ km (units: km).

| Dmr | Mean | STD | Median | IQR | Peak | Model |
|---|---|---|---|---|---|---|
| All | −1.08 | 2.48 | −0.65 | 2.19 | −0.38 | 1.84 |
| $H_m < 4$ km | −2.73 | 2.83 | −2.00 | 4.25 | −1.18 | 2.46 |
| $H_m > 6$ km | 0.03 | 1.58 | −0.17 | 1.39 | −0.25 | 1.06 |

## 4. Factors Affecting Retrieval Accuracy and Their Impacts

### 4.1. Influences from Cloud Properties

#### 4.1.1. D$_{mr}$ and Cloud Depth

When clouds are optically thick, they can be approximated as a black body ($E \approx 1$). The CTH retrieval for an optically thick cloud should be less complex than optically thin clouds ($E < 1$). The cloud optical thickness, which is determined by the cloud water content and particle size, is different from

the cloud depth. Although, statistically, optically thick clouds are generally geometrically thick too. The MODIS cloud optical thickness is not presently available for all comparisons, but cloud depth can be measured by KPDR. Therefore, cloud depth is used as a proxy for the cloud optical depth to investigate the impacts from cloud thickness. The relationship between $D_{mr}$ and cloud depth is shown in Figure 3.

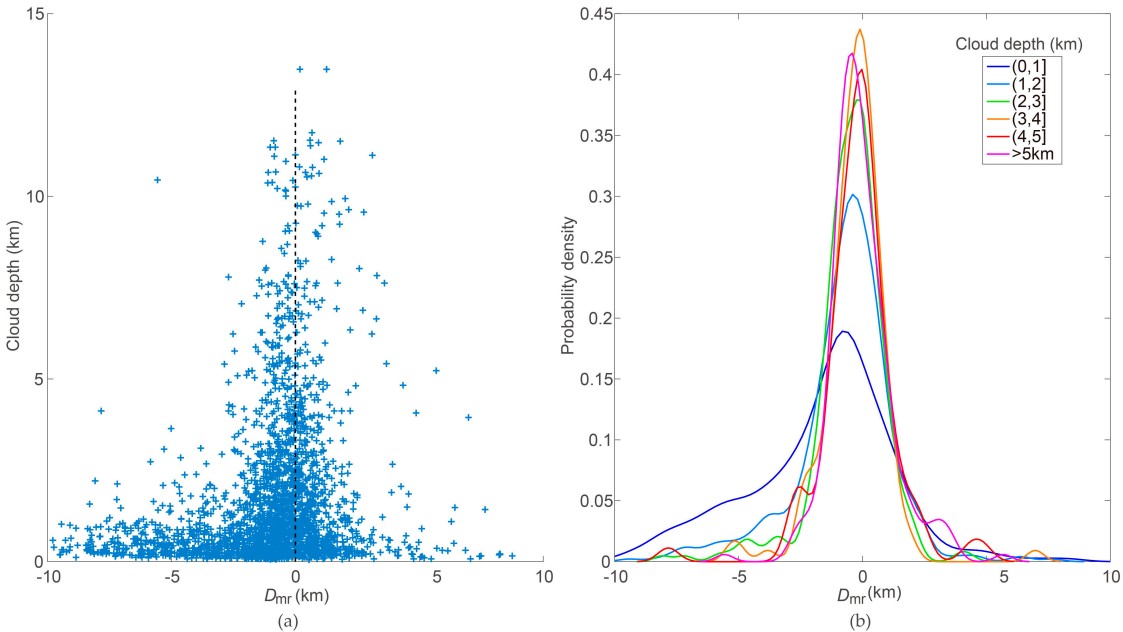

**Figure 3.** Relationship between $D_{mr}$ and cloud depth, (**a**) and individual distributions of $D_{mr}$ in terms of cloud depth (**b**).

It is apparent that the value of $D_{mr}$ relates closely to the cloud depth (Figure 3a). For thin clouds (cloud depth <1 km), $D_{mr}$ occurs between −12 and 9 km. However, when the clouds become thicker, especially larger than 2 km, the $D_{mr}$ range narrows considerably toward zero. For comparisons of cloud depth > 1 km (47% of all comparisons), the mean $D_{mr}$ is −0.47 ± 1.68 km. Table 3 shows the statistical results of $D_{mr}$ in different cloud depth ranges.

**Table 3.** $D_{mr}$ for different cloud depths (CD) (units: km).

| $D_{mr}$ | CD $\in$(0,1) | CD $\in$(1,2) | CD $\in$(2,3) | CD $\in$(3,4) | CD $\in$(4,5) | CD > 5 |
|---|---|---|---|---|---|---|
| Mean | −1.62 | −0.73 | −0.50 | −0.30 | −0.16 | −0.07 |
| STD | 2.92 | 2.00 | 1.45 | 1.30 | 1.53 | 1.29 |
| Median | −1.10 | −0.49 | −0.34 | −0.20 | −0.15 | −0.28 |
| IQR | 3.39 | 1.80 | 1.30 | 1.21 | 1.23 | 1.26 |
| Peak | −0.81 | −0.39 | −0.10 | −0.10 | −0.02 | −0.41 |
| Model | 2.45 | 1.44 | 1.01 | 0.86 | 0.98 | 0.92 |

Note that both the $CO_2$-slicing technique and IRW technique are based on the infrared radiative properties of the whole cloud layer. Normally, the infrared brightness temperature, measured by MODIS, is from the radiation center of the cloud layer, which might be different to the actual cloud top, as the radiation center is generally near to the central position of the cloud layer for optically thin clouds, whereas the radiation center of optically thick clouds is near the cloud top [14]. The retrieved $H_m$ of optically thin clouds is therefore lower than the actual CTH, even when other retrieval errors are not considered. In other words, relative to the actual CTH, the $H_m$ for optically thin clouds

may be underestimated to a larger extent than for optically thick clouds. However, this is not the decisive reason behind the dependence on cloud depth. Cloud depth determines cloud emissivity. Clouds are assumed to be black bodies, which is nearly correct for thick clouds, but less so for thin clouds. Due to more information from the surface obtained by MODIS, understanding the emissivity of optically thin clouds is difficult with respect to the retrieval algorithm, and poor estimation will lead to large uncertainties.

### 4.1.2. $D_{mr}$ and CBH

The CBH indicates to what level clouds belong. To reduce the impacts of cloud depth on $D_{mr}$, we analyze the $D_{mr}$ by dividing cloud depth into several groups: <1, 1–2, 2–3, 3–4, and ≥4 km. Figure 4b–f present the $D_{mr}$ changes as a function of CBH within the five cloud depth ranges. When the cloud depth is >2 km (Figure 4d–f), the $D_{mr}$ is independent of the CBH. When the cloud depth is < 1 km, the mean $D_{mr}$ gets lower as the CBH increases. One reason for this is that the CBH value limits the range of $D_{mr}$ variation when the cloud depth is restricted within an appointed range. That is, when the CBH is <3 km and the cloud depth is <1 km, the $H_r$ must be less than 4 km, and then the $D_{mr}$ should be no less than −4 km. Hence, the relationship between $D_{mr}$ and CBH is not confirmed, although the mean or median $D_{mr}$ changes as the CBH decreases. In Figure 4b, when the CBH is less than 1 km, positive $D_{mr}$ values are dominant, while negative $D_{mr}$ values appear dominantly when the CBH is larger than 5 km. There is a tendency for MODIS to underestimate high thin clouds, but overestimate low thin clouds. Overall, the relationship between $D_{mr}$ and CBH is unobvious from Figure 4.

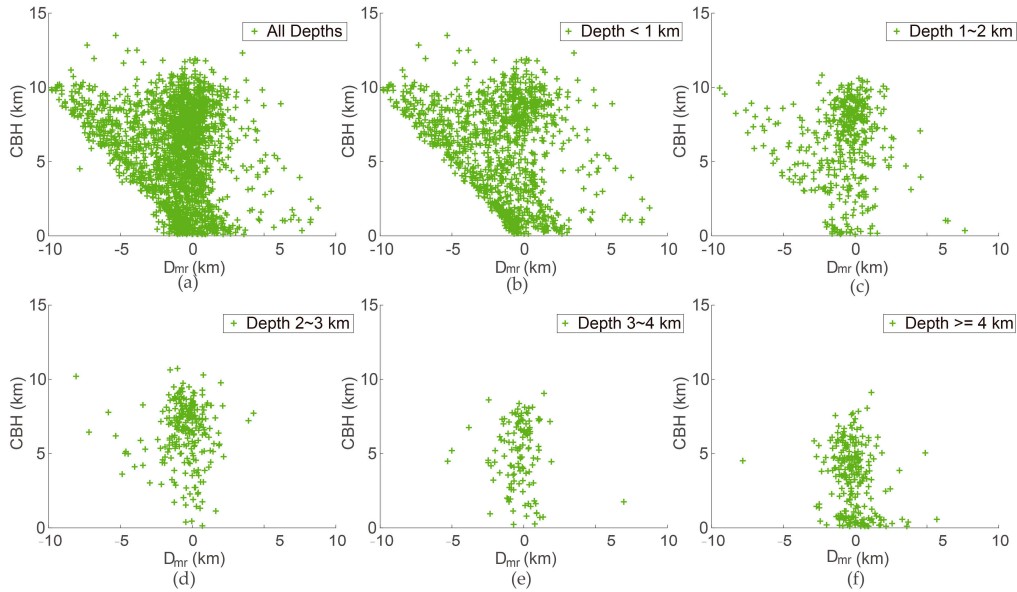

**Figure 4.** Relationship between $D_{mr}$ and CBH for all comparisons (**a**), for cloud depth <1 km (**b**), for cloud depth between 1 km and 2 km (**c**), for cloud depth between 2 km and 3 km (**d**), for cloud depth between 3 km and 4 km (**e**), and for cloud depth ≥4 km (**f**).

### 4.1.3. $D_{mr}$ and Cloud Fraction

Two cloud fraction datasets (MODIS cloud fraction and ERA5 cloud fraction) are used to investigate the relationship between cloud fraction and $D_{mr}$ (see Figure 5). Since large differences are mostly from clouds with low depth, only collocations with cloud depths of <2 km are presented in Figure 5. The number of accessible MODIS cloud fraction data is less than the ERA5 cloud fraction data, meaning there are fewer points in Figure 5a than in Figure 5b. In spite of this, the scatterplots based on the two datasets all show that $D_{mr}$ is not closely correlated with the cloud fraction, although statistical analysis based on both fraction datasets shows that the mean difference (−1.06 ± 2.50 km) for cloud fraction >50% is a little higher than the mean difference (−1.15 ± 2.63 km) for cloud fraction <50%.

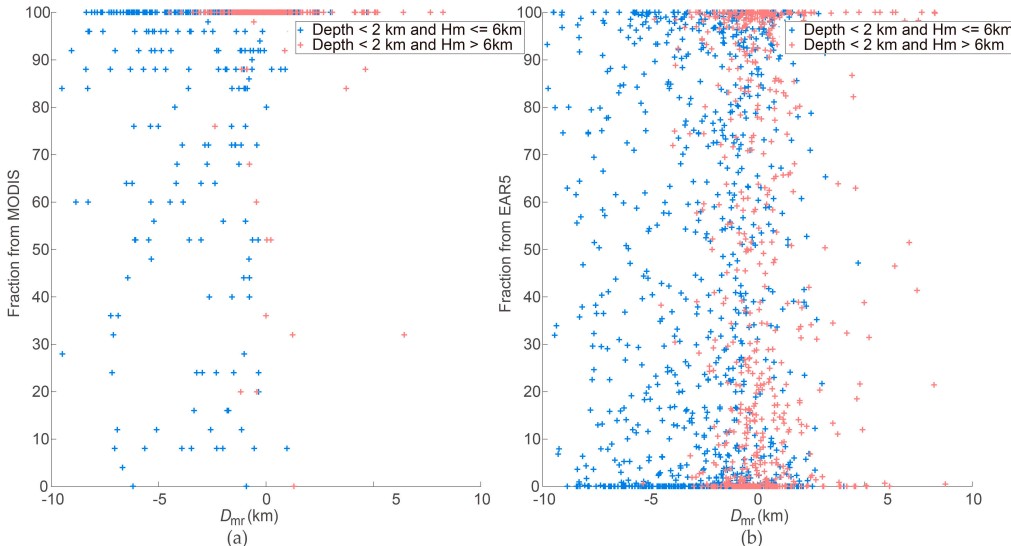

**Figure 5.** Relevance of MODIS cloud fraction, (**a**) and ERA5 cloud fraction, (**b**) in $D_{mr}$ from collocations with cloud depth <2 km. Collocations with cloud depth <2 km and $H_m$ >6 km are illustrated with red points.

### 4.1.4. $D_{mr}$ and Cloud Layer and Season

Two methods are used to compute the CTH of radar data for multi-layer cloud (see Section 2.2). The differences in CTH for cloud with different layers through different calculation methods are compared.

There are about 31% comparisons containing multi-layer clouds (cloud will be regarded as layered if the height gap between two cloudy bins is larger than 150 m). In Figure 6a, the distribution of $D_{mr}$ in terms of single layer and multi-layer cloud are presented and calculated using two calculation methods. For multi-layer cloud, it can be seen that the statistical results from two calculation methods are close to each other (see black dashed line and red line in Figure 6a). In addition, the discrepancy for multi-layer cloud is close to that of single-layer cloud in Beijing. Based on this statistic results, the comparisons in other Sections use the $H$ as a representative for the $H'$. Figure 6b presents the distribution of $D_{mr}$ in spring, summer, autumn and winter in Beijing. The mean $D_{mr}$ in winter is $-1.58$ km ($\pm 2.30$ km), which is the largest of the four seasons. The mean $D_{mr}$ in summer is $-0.89$ km ($\pm 2.78$ km), which is the smallest difference of the four seasons and lower than the average difference due to more thick clouds occurring in summer. Summer in Beijing has the largest STD due to more diverse distribution of cloud compared with the other seasons.

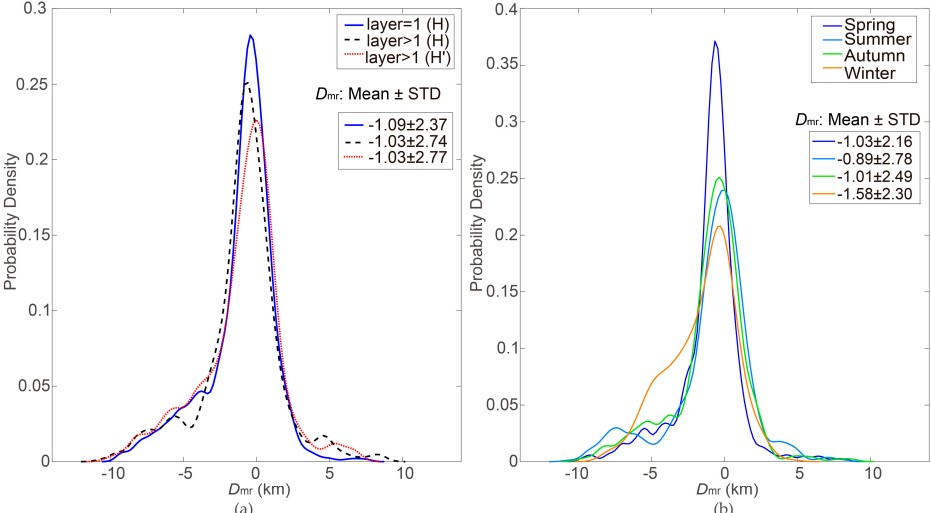

**Figure 6.** The distribution of $D_{mr}$ in terms of cloud layer number with two computation method (**a**) and season (**b**). In (**a**), "layer >1" means multi-layer cloud.

### 4.2. Influences from Atmospheric Parameters

Calculations of clear-sky radiances for each MODIS band play crucial roles in the CTH retrieval algorithm. For the IRW technique, clear-sky radiances are used as a reference to determine the CTH. For the $CO_2$-slicing technique, the clear-sky radiance affects the radiance ratio (observed radiance minus clear-sky radiance). The MODIS clear-sky radiances are estimated by a radiative transfer model named PFAAST and biases of input atmospheric parameters to actual parameters may lead to computation uncertainty, resulting in retrieval uncertainty. Emphasis of this Section is on investigation of the influences from atmospheric parameters on the CTH difference.

#### 4.2.1. $D_{mr}$, TWV and Ozone

TWV and ozone are two important parameters determining clear-sky infrared temperature brightness due to their absorptions in infrared band. Collocations with depths of <2 km have relatively large $D_{mr}$ and are compared to investigate the possible influences. As shown in Figure 7a, the points are scattered "everywhere" and $D_{mr}$ is not sensitive to the TWV. Here, the relevance of TWV in $D_{mr}$ is unobvious. In Figure 7b, the $D_{mr}$ values get slowly closer to zero as total ozone gets higher, and there is a weak decline in the tendency of the discrepancy as total ozone increases.

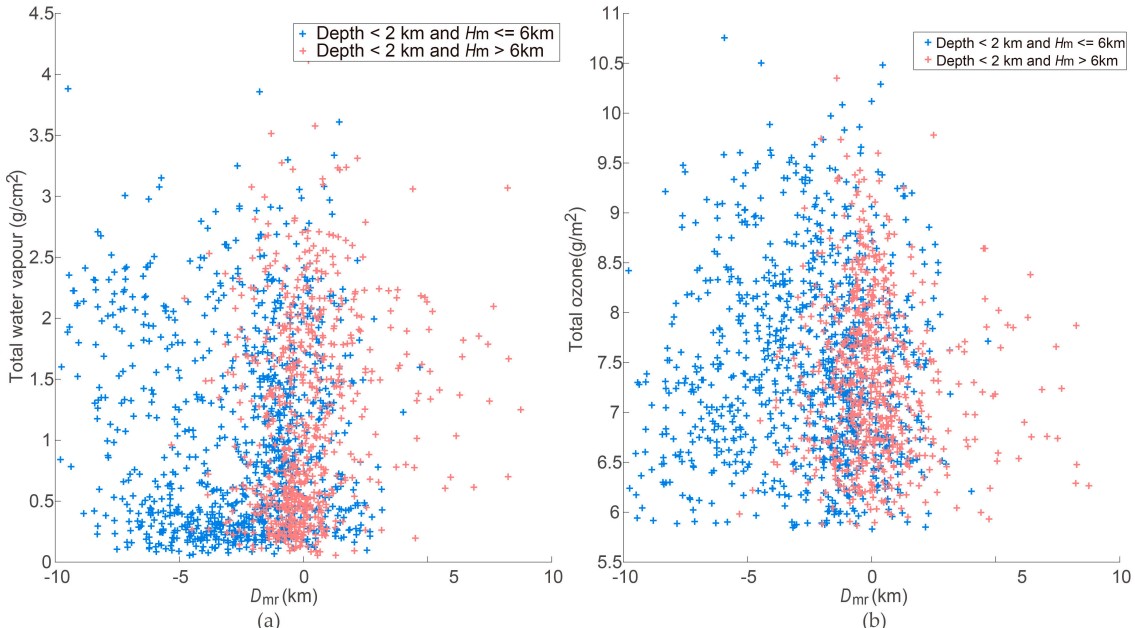

**Figure 7.** Relationship of $D_{mr}$ with TWV (**a**) and total ozone (**b**). Collocations illustrated here are the same as in Figure 5.

#### 4.2.2. $D_{mr}$ and Temperature Inversion

The IRW technique retrieves the CTH by comparing the measured brightness temperature with the calculated vertical brightness temperature profile produced from PFAAST. When temperature inversion is present, multiple matches might be found, and then biases will be produced. In this Section, two types of datasets, with temperature inversion and without temperature inversion, are separated and compared to investigate the relevance of temperature inversion in the $D_{mr}$ (Figure 8). Large $D_{mr}$ values appear in both datasets and a distinct difference between the two datasets is not found. In the presence of temperature inversion, large $D_{mr}$ occurs at different inversion heights and lapse rates. Statistical analysis shows that $D_{mr}$ is not sensitive to temperature inversion over Beijing.

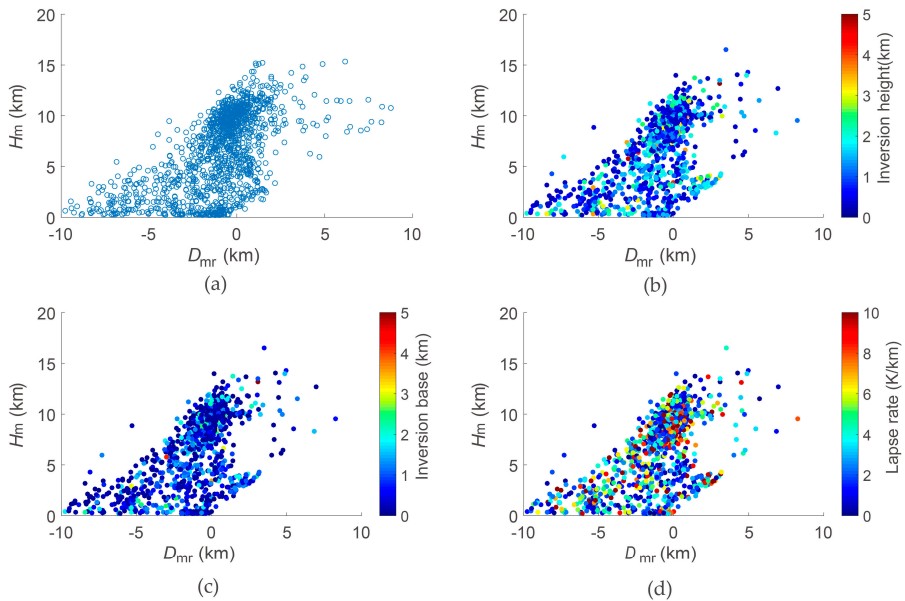

**Figure 8.** Relevance of temperature inversion in the $D_{mr}$. The $D_{mr}$ values from collocations without temperature inversion are shown in (**a**). In the presence of temperature inversion, (**b**) shows the relationship between $D_{mr}$ and temperature inversion height, (**c**) is the relationship between $D_{mr}$ and the temperature inversion base, and (**d**) illustrates the relationship between $D_{mr}$ and a negative temperature lapse rate.

### 4.2.3. $D_{mr}$ and Surface Atmospheric Properties

An abnormal atmospheric status, such as severe air pollution or dust, may lead to considerable bias between the measured brightness temperature and calculated brightness temperature because input atmospheric profiles for abnormal weather may not be adjusted appropriately in time for model calculation. Synergistic surface observations of $PM_{10}$, CO, $NO_2$ and $SO_2$ are applied in this Section to investigate their impacts on CTH retrieval (Figure 9). For $PM_{10}$ particles, the average difference has a weak decrease as the concentration increases. Similar results are found for CO, $NO_2$ and $SO_2$. The Pearson correlation coefficients between the $PM_{10}$, $SO_2$, CO, $NO_2$ and the $D_{mr}$ for cloud with depth < 1 km and $H_m$ >6 km are −0.13, −0.13, −0.14, and −0.13, respectively. Therefore, it can be concluded that air pollution or dust are not crucial sources of large $D_{mr}$.

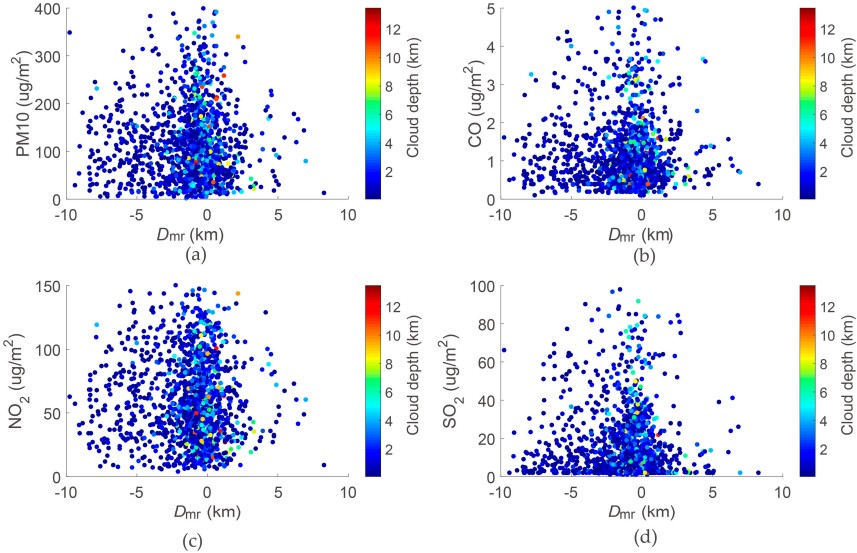

**Figure 9.** Relevance of $PM_{10}$ (**a**), CO (**b**), $NO_2$ (**c**) and $SO_2$ (**d**) in $D_{mr}$. The color bar in each panel indicates the cloud depth.

## 4.3. $D_{mr}$, Retrieval Method and Viewing Angle

Figure 10a presents all $D_{mr}$ values retrieved from the IRW technique and $CO_2$-slicing technique with individual colors. It is found that the $CO_2$-slicing technique performs much better than the IRW technique for mid- and high-level clouds (CBH >2 km), while IRW technique is better for low-level cloud (CBH ≤2 km). Most large differences are produced by the IRW technique. Table 4 presents the statistical results of the two retrieval methods. The mean $D_{mr}$ for the $CO_2$-slicing technique is 0.09 km, whereas it is −2.20 km for the IRW technique. The $CO_2$-slicing technique shows a weak tendency of overestimation for clouds when the CBH is less than 2 km. In Section 4.1.2, it is found that MODIS tends to underestimate high thin cloud but overestimate low thin clouds, which is also shown in Figure 10a. When the CBH is less than 1 km, the IRW technique tends to overestimate the CTH (0.04 ± 1.29 km). However, when the CBH is above 4 km, most CTHs are underestimated by the IRW technique (−4.14 ± 2.60 km). Overall, the IRW technique produces more large uncertainties.

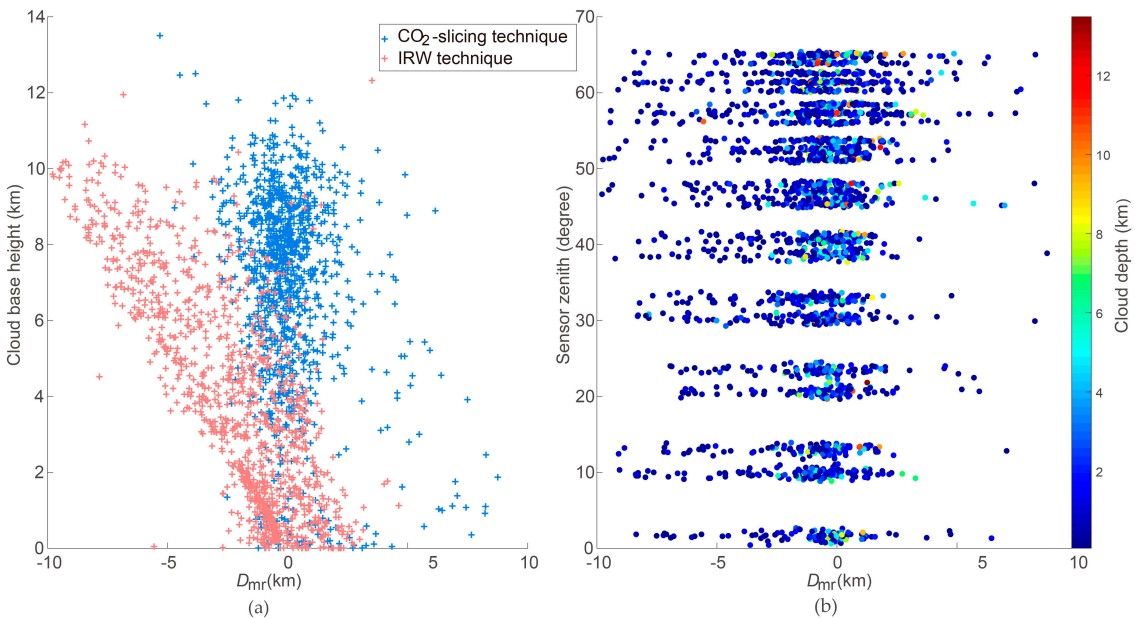

**Figure 10.** (**a**) Distribution of $D_{mr}$ calculated from two retrieval techniques: $CO_2$-slicing technique (blue); IRW technique (red). (**b**) Relationship between $D_{mr}$ and MODIS viewing zenith angle.

**Table 4.** Statistics of the $D_{mr}$ retrieved by the $CO_2$-slicing technique and IRW technique (units: km).

| $D_{mr}$ | Mean | STD | Median | IQR | Peak | Model |
|---|---|---|---|---|---|---|
| $CO_2$-slicing | 0.09 | 1.58 | −0.13 | 1.40 | −0.25 | 1.19 |
| IRW | −2.20 | 2.73 | −1.58 | 3.62 | −0.95 | 2.33 |

The MODIS viewing geometry changes during each overpass over Beijing, resulting in various measuring paths that may cause uncertainties in CTH retrieval. Figure 10b shows the relationship between the $D_{mr}$ and sensor viewing angle (sensor zenith angle). The mean $D_{mr}$ increases as the sensor zenith angle increases, due to the longer measuring path and changing viewing geometry. However, the impacts of viewing angles are also associated with cloud depth. For thick cloud, what the MODIS measures can be regarded as the radiance from the cloud. For optically thin cloud, radiance that the MODIS measures is from the cloud, the atmosphere and the surface together. As the viewing angle increases, the radiance obtained by MODIS may be from longer radiative path. Then, more uncertainties will be produced. Therefore, the CTH retrieved for thin clouds is more sensitive to the viewing angle than for thick clouds.

## 5. Discussion

Knowledge of the emissivity of clouds, especially for thin clouds, is a challenge for the CTH retrieval algorithm and leads to large uncertainties since clouds are assumed to be black bodies, which is relatively accurate for thick clouds, but less so for thin clouds. The results found in this paper further reveal the potential limitations of passive infrared remote sensing on the CTH; that is, the black-body cloud assumption is conditional and restricts the retrieval accuracy. The $CO_2$-slicing technique is a better solution for effective emissivity estimation, resulting in good accuracy. However, for the IRW technique, it is likely that poor estimations of cloud effective emissivity lead to large uncertainties.

## 6. Conclusions

MODIS CTH products have been widely applied in the meteorological community since the instrument's launch in 2000. This paper investigates the accuracy of MODIS CTH by comparisons with measurements from a ground-based Ka band radar over Beijing. Thorough investigation is a complex task, since the retrieval accuracy depends on the instrument's calibration, the properties of the spectral response functions, the atmospheric profiles, cloud identification, the calculation of the radiance at the top of the atmosphere. The focus of this paper is to study the distribution of the difference, and its relationship with cloud properties and atmospheric properties, not including the instrument's observational properties.

It is found that MODIS underestimates the CTH by, on average, $-1.08 \pm 2.48$ km, relative to the radar data. The discrepancy is dominantly and strongly associated with the cloud depth. Statistical analysis shows that the differences decline as the cloud depth increases. It is found that atmospheric parameters causing potential calculation errors have few relevance to the discrepancy. The large differences arise primarily from the retrieval of CTH for thin and high-level clouds by the IRW technique, for example, the mean difference is $-4.86 \pm 2.49$ km when CBH $\geq 5$ km and CD $\leq 2$ km. On the contrary, the $CO_2$-slicing technique illustrates a better performance. For thin clouds, MODIS tends to underestimate high-level clouds, but overestimate low-level clouds. Some of these qualitative conclusions have also been confirmed in several previous comparison work. However, the quantitative results, presented in this paper, for Beijing area are more specific.

As addressed above, the accuracy of the MODIS CTH retrieval algorithm is affected by multiple factors; for instance, theoretical assumptions, the instrument calibration for radiance observation, the instrument spectral response function, the sensor viewing angle, the atmospheric description, the cloud mask and the calculation of top-of-the-atmosphere radiance. A shortage of information on the instrument's status and model simulations limits a complete analysis of the causes of differences. Our statistical analysis, based on four years of ground-based KPDR measurements, provides guidance for the application of MODIS CTH products and a reference for improvements in the retrieval algorithm. We hope that further analysis will be performed in the future when more auxiliary data are accessible.

**Author Contributions:** Conceptualization, J.H. and D.L; methodology, J.H.; software, J.H. and C.H.; validation, J.H., Y.B. and J.L.; formal analysis, J.H.; investigation, J.H.; resources, J.L.; data curation, Y.B. and J.L.; writing—original draft preparation, J.H.; writing—review and editing, J.H.; visualization, M.D.; supervision, M.D.; project administration, D.L.; funding acquisition, J.H. All authors have read and agreed to the published version of the manuscript.

**Funding:** This work was funded by the National Natural Science Foundation of China (grants 41775032 and 41275040).

**Acknowledgments:** We appreciate the NASA Aqua/Terra MODIS team and ECMWF ERA5 science team for generously sharing those data. We thank the many contributors from the Ka band radar science team of the IAP who enabled our research and made this project possible.

**Conflicts of Interest:** The authors declare that they have no known competing financial interests or personal relationships that could have appeared to influence the work reported in this paper.

**Appendix A**

A detailed description of the scheme developed to collocate MODIS CTH data with KPDR data is presented.

The time that KPDR requires to scan a certain cloud path depends on the speed of the moving cloud. If the speed of a moving cloud is 10 km/h (roughly the speed of a light breeze), KPDR takes about 6 min to scan 1 km and about 30 min to scan 5 km. If the wind speed is 36 km/h, then KPDR will scan 36 km in 1 h. MODIS CTH data around KPDR also cover various spatial areas depending on the viewing geometry of each overpass. Since both datasets show various temporal and spatial resolutions, we try several data collocation methods using different KPDR observation intervals and MODIS spatial areas to compare and find the optimal collocation scheme.

KPDR is stationary and its location is used as a standard for the preliminary selection of MODIS data. MODIS CTH is computed by three different methods: (1) MODIS CTH is the CTH of the MODIS pixel nearest to KPDR (referred to as $H_{m0}$); (2) MODIS CTH is the average CTH of MODIS pixels within 5 km to KPDR (referred to as $H_{m5}$); and (3) MODIS CTH is the average CTH of MODIS pixels within 30 km to KPDR (referred to as $H_{m30}$).

The observation time ($T_{m0}$) of the nearest MODIS pixel is used as a reference for selection of KPDR data. KPDR CTH is calculated based on two time intervals; one is averaged based on all cloudy profiles occurring within $T_{m0} \pm 5$ min (termed $H_{r5}$); and the other is averaged within 30 min ($T_{m0} \pm 30$ m, termed $H_{r30}$).

We perform four collocations: (1) $H_{m0}$ versus $H_{r5}$; (2) $H_{m5}$ versus $H_{r5}$; (3) $H_{m5}$ versus $H_{r30}$; and (4) $H_{m30}$ versus $H_{r30}$. From 1 January 2014 to 31 December 2017 there are 2241 comparisons for the collocation of $H_{m0}$ versus $H_{r5}$, 2317 comparisons for $H_{m5}$ versus $H_{r5}$, 2522 comparisons for $H_{m5}$ versus $H_{r30}$, and 2888 comparisons for $H_{m30}$ versus $H_{r30}$. Figure A1 shows the distribution of the differences using the four collocation schemes, and the quantified differences are presented in Table A1. Overall, the collocation scheme of $H_{m5}$ versus $H_{r5}$ shows a lower mean, median and peak difference among the four collocation schemes. In addition, the $D_{m5r5}$ (equal to $H_{m5}$ minus $H_{r5}$) is close to the average difference of the four collocation schemes. Therefore, the scheme of $H_{m5}$ versus $H_{r5}$ is used in this paper for data collocation.

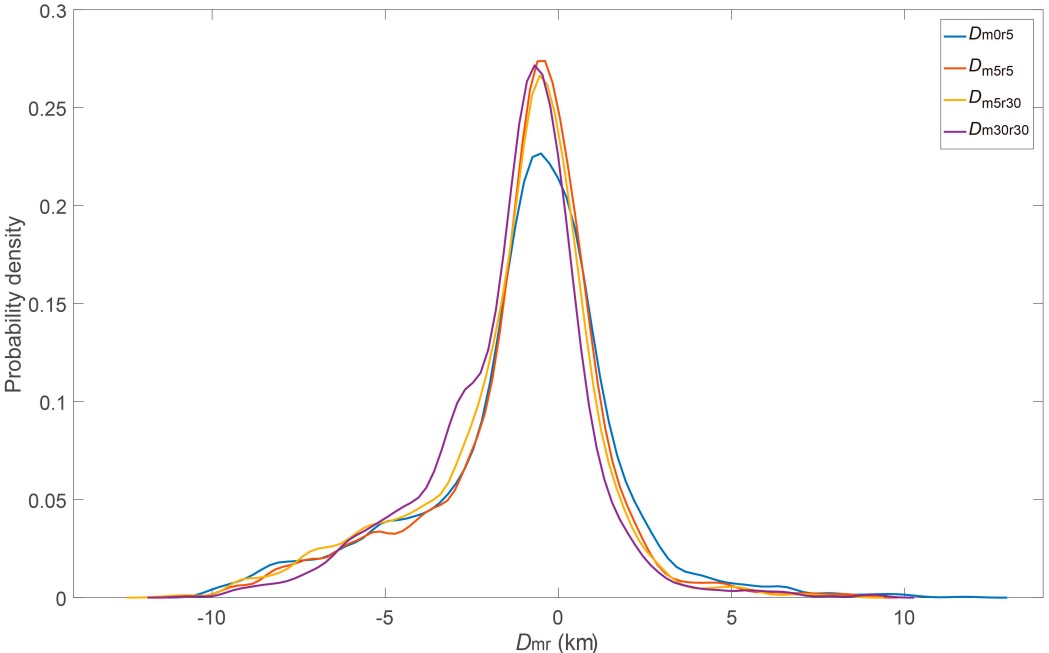

**Figure A1.** Distributions of the probability density of $D_{mr}$ using four collocation schemes ($D_{m0r5}$: $H_{m0}$ versus $H_{r5}$; $D_{m5r5}$: $H_{m5}$ versus $H_{r5}$; $D_{m5r30}$: $H_{m5}$ versus $H_{r30}$; $D_{m30r30}$: $H_{m30}$ versus $H_{r30}$.

**Table A1.** Statistics of the CTH difference between MODIS and KPDR using four collocation schemes (units: km).

| Scheme | Mean | STD | Median | IQR | Peak | Model |
|---|---|---|---|---|---|---|
| $D_{\text{m0r5}}$ | −1.018 | 2.782 | −0.670 | 2.195 | −0.501 | 1.836 |
| $D_{\text{m5r5}}$ | −1.076 | 2.484 | −0.645 | 2.510 | −0.384 | 2.051 |
| $D_{\text{m5r30}}$ | −1.305 | 2.506 | −0.760 | 2.419 | −0.537 | 1.945 |
| $D_{\text{m30r30}}$ | −1.343 | 2.293 | −0.958 | 2.436 | −0.683 | 1.897 |
| Average | −1.198 | 2.511 | −0.770 | 2.417 | −0.701 | 1.929 |

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
