# Peer review of "Measurement of Cloud Top Height: Comparison of MODIS and Ground-Based Millimeter Radar"

_remotesensing, doi:10.3390/rs12101616_

Round 1

Reviewer 1 Report

This manuscript presents a comparison of 4 years of MODIS CTH products with CTH measurements from a Ka band radar located in Beijing. Most results are presented in terms of the distribution of the difference between the MODIS CTH and radar CTH and corresponding statistics. The results are stratified such that the impact of various cloud and atmospheric properties on the CTH difference could be analyzed in more detail.

The study finds, on average, an underestimation of the MODIS CTH with respect to radar by about 1 km. This difference is smaller for higher clouds and larger for lower clouds. The cloud depth shows a strong impact on the CTH difference, while other cloud and atmospheric properties only show a weak or no impact. The CO2-slicing technique performs better than the IRW technique.

I find the study relevant since CTH products from passive imagers are used frequently in climate studies and specifically the MODIS cloud products are widely used in the research community. A number of studies already exist that evaluate the MODIS cloud products in much detail, also with comparison to radar measurements, and in that sense this study is not a novelty. However, on-going evaluation studies are needed, especially after retrieval updates as well as for continued improved understanding and quantification of the performance of MODIS CTH under a large variety of conditions.

Generally, the manuscript is well-structured and results are mostly clearly presented and discussed. At some points it would benefit from a bit more clarification and discussion as well as some minor text adjustments, see comments in attached pdf file. Therefore, I recommend minor revision.

Reviewer 2 Report

This nice paper presents an analysis of the comparison of cloud top height from MODIS satellite measurements and from Ground-based millimeter Radar measurements. The paper is well written, and the analysis of the reasons of the discrepancies is well conducted. The authors have identified and quantified all the sources that can produce such discrepancies.

The paper can be published as it is, in the authors consider this small comment: Line 285-286 is unclear (H” for representative of H’).

Author Response

Dear Reviewer,

We thank you very much for your comments.

We delete the quotation mark in the revised paper.

Reviewer 3 Report

Review of the study entitled “Measurement of Cloud Top Height: Comparison of MODIS and Ground-based Millimeter Radar” by Juan Huo et al.

The study compares cloud top height derived by MODIS with ground-based radar measurements over Beijing. The satellite retrievals by MODIS are shown to underestimate cloud top height around Beijing. These underestimations are shown to be dependent on the retrieval method. The CO2-slicing technique is shown to perform better than the 11-μm infrared window technique compared to the radar measurements. There is also dependence on height. It is shown that MODIS cloud top heights greater than 6 km agree better with radar data that cloud top heights less than 4 km. A little dependence on air pollution is revealed. The study is well written and I recommend publication after few clarifications and corrections.

1) Figure 1 caption: Mention that the figure refers to Beijing.

2) l.201: you write “non-Gaussian distribution”. To my eyes it looks Gaussian distribution, maybe not a perfect one but almost Gaussian.

3) l.240: I cannot what the authors mean with the line “Due to more information from the surface obtained by MODIS”.

4) l.284: It is black dashed line not blue dashed line.

5) l.332: The authors state that “for NO2 such a weak relationship is not apparent” but a weak relationship is shown as in all others.

6) All sections: It is not clear which MODIS retrieval method is used in section 3 as well as in all sections before section 4.3. I assume that it is a combination of the CO2-slicing technique and the IRW technique. If so, how would the results change in all sections if data only from the CO2-slicing technique were used? Would there be better relationships? This should be clarified since the authors come up with the finding that the CO2-slicing technique produces better agreement between the satellite and the radar data.

Author Response

Dear Reviewer,

Thank you very much for your comments and suggestions. According to your suggestions, we have made careful revisions which are marked using the “Track Changes” function. Our responses (blue) to your comments (black) are as follows.

General comments:

The study compares cloud top height derived by MODIS with ground-based radar measurements over Beijing. The satellite retrievals by MODIS are shown to underestimate cloud top height around Beijing. These underestimations are shown to be dependent on the retrieval method. The CO2-slicing technique is shown to perform better than the 11-μm infrared window technique compared to the radar measurements. There is also dependence on height. It is shown that MODIS cloud top heights greater than 6 km agree better with radar data that cloud top heights less than 4 km. A little dependence on air pollution is revealed. The study is well written and I recommend publication after few clarifications and corrections.

Specific comments:

1) Figure 1 caption: Mention that the figure refers to Beijing.

“Beijing” has been added in the caption of Figure1.

2) l.201: you write “non-Gaussian distribution”. To my eyes it looks Gaussian distribution, maybe not a perfect one but almost Gaussian.

We regard it as “non-Gaussian distribution” because the peak center of the distribution is not at the value of zero and data with negative values have more than that with positive values. For a perfect Gaussian distribution, the median value is generally equal to the mean value. Therefore, two statistical methods (mean value and median value) are used to calculate the difference to make the results more clearly.

3) l.240: I cannot what the authors mean with the line “Due to more information from the surface obtained by MODIS”.

It means that infrared radiance collected by MODIS for optically thin cloud also may include the radiance from the surface and the atmosphere below cloud. For thick cloud, infrared radiance collected by MODIS is the radiance of cloud because the radiation from surface or atmosphere are “blocked” or “absorbed” by thick cloud and cannot reach MODIS.

4) l.284: It is black dashed line not blue dashed line.

Sorry for that. It has been corrected.

5) l.332: The authors state that “for NO2 such a weak relationship is not apparent” but a weak relationship is shown as in all others.

These statements are obtained from the distribution feature in Figure 9. To understand their correlation, we calculated the Pearson correlation coefficient between the PM10, SO2, CO, NO2, total ozone, total water vapor, temperature inversion and the Dmr for clouds with depth < 1 km and Hm > 6 km. The correlation coefficients are -0.13, -0.13, -0.14, -0.13, -0.10, 0.16, and 0.20, respectively. Then, there should be no correlation for all factors if 0.3 is used as threshold to determine whether they are relevant. Based on the calculation results, some previous conclusions should be corrected. We revised the statements in this section, the abstract and summary section.

6) All sections: It is not clear which MODIS retrieval method is used in section 3 as well as in all sections before section 4.3. I assume that it is a combination of the CO2-slicing technique and the IRW technique. If so, how would the results change in all sections if data only from the CO2-slicing technique were used? Would there be better relationships? This should be clarified since the authors come up with the finding that the CO2-slicing technique produces better agreement between the satellite and the radar data.

The MODIS CTH are retrieved by the combination of CO2-slicing technique and the IRW technique, namely, some CTHs are derived by IRW technique while others are derived by CO2-slicing technique. There are some clouds for which CO2-slicing technique is not applicable. Section 3 presents a general knowledge of the difference between MODIS CTH and radar CTH to readers and potential users. It is necessary and meaningful because the users do not know which retrieving technique is used for each MODIS CTH when they use the data. Statistical results, excluding comparisons retrieved by IRW technique, are incomplete and one-sided. Section 4.3 presents the accuracy of the two techniques in detail, which helps us understand the performances of the two techniques. Therefore, it might be better to keep those analysis unchanged. According to the comments from other reviewers, we made a few revisions in the file about the comparisons between the two techniques and the analysis results should be presented more clearly.

Round 2

Reviewer 3 Report

Tha study has been revised and can be published.